# Epidemiology of Stroke in Sickle Cell Disease

**DOI:** 10.3390/jcm10184232

**Published:** 2021-09-18

**Authors:** Fenella Jane Kirkham, Ikeoluwa A. Lagunju

**Affiliations:** 1Developmental Neurosciences, UCL Great Ormond Street Institute of Child Health, Faculty of Population Health Sciences, 30 Guilford Street, London WC1N 1EH, UK; 2Child Health, Clinical and Experimental Sciences, Faculty of Medicine, University of Southampton and University Hospital Southampton, Southampton SO16 6YD, UK; 3Paediatric Neurosciences, King’s College Hospital, London SE5 9RS, UK; 4Department of Paediatrics, College of Medicine, University of Ibadan, Ibadan PMB 3017, Nigeria; ilagunju@yahoo.co.uk; 5Department of Paediatrics, University College Hospital, Ibadan PMB 5116, Nigeria

**Keywords:** anemia, sickle cell, cerebrovascular disorders, incidence, prevalence, stroke, intracranial hemorrhage, seizures, epilepsy, headache, cognition, paraplegia, neuropathy, myopathy

## Abstract

Sickle cell disease is the most common cause of stroke in childhood, both ischaemic and haemorrhagic, and it also affects adults with the condition. Without any screening or preventative treatment, the incidence appears to fall within the range 0.5 to 0.9 per 100 patient years of observation. Newborn screening with Penicillin prophylaxis and vaccination leading to reduced bacterial infection may have reduced the incidence, alongside increasing hydroxyurea prescription. Transcranial Doppler screening and prophylactic chronic transfusion for at least an initial year has reduced the incidence of stroke by up to 10-fold in children with time averaged mean of the maximum velocity >200 cm/s. Hydroxyurea also appears to reduce the incidence of first stroke to a similar extent in the same group but the optimal dose remains controversial. The prevention of haemorrhagic stroke at all ages and ischaemic stroke in adults has not yet received the same degree of attention. Although there are fewer studies, silent cerebral infarction on magnetic resonance imaging (MRI), and other neurological conditions, including headache, epilepsy and cognitive dysfunction, are also more prevalent in sickle cell disease compared with age matched controls. Clinical, neuropsychological and quantitative MRI screening may prove useful for understanding epidemiology and aetiology.

## 1. Introduction

Although the sickle gene mutation is most common in equatorial Africa, it is also found in the Mediterranean regions of Europe and Turkey, related to the distribution of malaria [1], and sickle cell disease (SCD) is now distributed worldwide as a consequence of the slave trade and economic migration. Stroke is a well-recognised complication of homozygous sickle cell anaemia (SCA; haemoglobin SS-HbSS) [2] and also occurs in compound heterozygotes, e.g., those with HbSC disease and HbSβthalassaemia (HbSβthal) [2,3].

There are data on the epidemiology of stroke in SCD from cohort studies, records of patients enrolled in healthcare provision, discharge data from hospitalisations and treatment trials. A major difficulty is the lack of information about diagnostic criteria for inclusion. Earlier studies used definitions which were not necessarily standardised [4] and, although more recent studies have included neuroimaging, as well as using the World Health Organisation definition of stroke, there is little evidence of neurological or neuroradiological over-reading or of the inclusion/exclusion criteria applied. In fact, there is a wide differential of focal and generalised vascular and non-vascular pathologies [5,6], often distinguished using acute MR techniques [7], with important management implications.

Acute neurological symptoms and signs are common in SCD and, as well as stroke, include transient ischaemic attack (TIA) [8], headaches [9,10], seizures [8,11,12] and coma [13]. Altered mental status with or without reduced level of consciousness, headache, seizures, visual loss or focal signs can occur in numerous contexts, including infection [14], acute chest syndrome (ACS) [15], acute anaemia [16], after surgery [17], transfusion [18] or immunosuppression [19] and apparently spontaneously [6]. For example, in one large series of patients with SCD and ACS, 3% of patients had neurological symptoms at presentation, and these symptoms developed in a further 7–10% as a complication of ACS [20]. These patients should be classified clinically as having had a cerebrovascular accident [2], although this may not occur during coding with the International Classification of Diseases.

## 2. Stroke

### 2.1. Definition and Overview

The World Health Organisation definition of clinical stroke is a focal neurological deficit lasting more than 24 h. Focal neurological deficits lasting less than 24 h are clinically termed TIAs, although acute neuroimaging may demonstrate abnormality, e.g., diffusion-weighted changes in focal ischaemia or subarachnoid haemorrhage in acute headache. Cerebral infarction may be symptomatic or asymptomatic (silent or covert infarct) [21]. In regions where the condition is prevalent, including Africa and parts of the USA and Europe, SCD is typically the most common cause of overt stroke in children [22,23,24]. Adults with SCD are also affected, with rising prevalence and incidence with age [2,25,26]. In the absence of screening and prophylactic treatment, between 5 and 17% of SCD patients will suffer a first stroke during childhood or adolescence [2,27,28,29,30,31,32,33], with a quarter affected by their 45th birthday [2]. In patients with SCD, the lifetime risk of overt stroke is between 25 and 30% [2].

### 2.2. Mortality of Stroke in Sickle Cell Disease

Stroke is an important cause of death in SCD, accounting for 10% of cases in the autopsy series covering the years 1929–1996 [34], although this reduced to 4% for the Howard University series from 1976 to 2001 [35]. Ischaemic stroke was not associated with death in the Co-operative study of Sickle Cell Disease (CSSCD), but in the series of hospitalised Californian patients with ischaemic stroke, 7% died [25], while 30% and 26% of patients with SCD and haemorrhagic stroke died in this study [25] and the CSSCD [2] respectively. Overall, 5% of children with SCD hospitalised for first or recurrent stroke died in the Californian study [25].

### 2.3. Incidence of All Stroke Presenting Clinically in Sickle Cell Disease without Prophylaxis

In childhood, early studies found that approximately 75% of the strokes were ischaemic and the remainder were haemorrhagic [27], but more recent data suggest that the proportion of ischaemic stroke is higher [36]. For patients of all genotypes not included in newborn screening but born after 1974 and recruited to the East London cohort [37] followed until 1998, the overall incidence of stroke was 0.67/100 PYO (Figure 1A), 0.60/100 PYO for ischaemic stroke and 0.07/100 PYO for haemorrhagic stroke. Importantly, with modern neuroimaging for screening as well as diagnosis, there appears to be considerable overlap in risk factors for ischaemic and haemorrhagic stroke [38]. In a North American study in Baltimore-Washington from 1988 to 1991, the overall incidence of childhood stroke was 0.0129/100 patient years of observation (PYO) (95% confidence intervals 0.0083–0.0211), while for ischaemic and haemorrhagic stroke respectively, incidences were 0.0058 (0.0037–0.0134) and 0.0071 (0.0026–0.0117) per 100 PYO. The most common cause was SCD (39%), with an incidence for first stroke of 0.285/100 PYO [0.105–0.622], 0.238 [0.078–0.556] and 0.0475 [0.012–0.266] per 100 PYO for ischaemic and haemorrhagic stroke respectively [24]. Children with SCD had a 221-fold increase in incidence for all strokes, 41-fold for ischaemic and 7-fold for haemorrhagic stroke. There is a greater proportion of haemorrhagic strokes in adulthood [2,25,30] and, as intracranial bleeding may be a cause of sudden death out-of-hospital [25], the incidence of haemorrhagic stroke may be underestimated.

### 2.4. Prevalence of All Stroke Presenting Clinically in Sickle Cell Disease without Prophylaxis

At entry to the CSSCD, the prevalence of cerebrovascular accident in all forms of SCD (HbSS, HbSC, HbSβthalassaemia) was 3.75% overall, ranging from 0.11% in those < 2 years old to 7.62% of those aged 40–49 years [2]. In a Brazilian study of 411 children, the overall prevalence of overt stroke was 5.1% (3–7.2%) [39]. In France the prevalence was 3.2% in those with HbSS, 1.2% in HbSC patients and 3.8% in HbSβthalassaemia [40]. At enrolment to the German registry, the prevalence of stroke was 4.2% overall, 5% in those with HbSS [41], while it was 4.5% (1/22) in a small Spanish series from 1985 to 2001 [42], reducing to 2.6% on entry to a larger later Spanish registry [43]. At baseline in 1991 in the East London cohort study [38], the prevalence of overt stroke was 3.8% (6 of 160). A 10 year study in England from January 2009 to December 2018 which involved 9503 National Health Service patients with SCD, median age 21 (interquartile range 9–34) years, excluding those born after 2008 and those who had had a stem cell transplant, found an overall rate of 3.8% for stroke and 8.1% for cerebrovascular symptoms [44].

There have been a number of studies reporting stroke prevalence in patients with SCD living in African countries (Table 1). Several of the studies are from various cities in Nigeria. Lagunju et al. undertook an initial study in Ibadan between 2004 and 2008 and found a prevalence of 6.8% (24/351) [31]. In a further study using the WHO criteria for stroke and TIA, with CT scan for all patients she found a prevalence of 8.4% [45]. Interestingly, in a case control study in Nigeria, paraplegia and stroke were both more common in children, adolescents and adults than in healthy controls [46]; it is not clear whether the paraplegias were related to prematurity, spinal stroke or acquired disease of the white matter. A meta-analysis of 30 studies in Africa, including all age groups, reported an overall prevalence for stroke of 4.2%, with a higher prevalence of 6.4% in studies using the WHO definition with neuroimaging, compared with 2.2% of those not clearly described [4]. In another systematic review and meta-analysis of 10 cross-sectional clinic-based studies in children with SCD living in Africa, prevalence ranged from 2.9 to 16.9%, leading to an estimate of 30,000 to 60,000 children affected [47].

Stroke appears to be less common in the Gulf states, with only 4 of 222 patients (1.8%) affected in a recent study from three of the four Gulf countries (Oman, Qatar, United Arab Emirates) with relatively high prevalences of homozygous sickle cell disease [48]. The prevalence of stroke also appears to be low in Kuwait, 1.4% in children and 2.3% in adults, [49] and in the population of children with SCD living in Shiraz in Iran [50]. One paediatric study from high and low altitude regions of SouthWest Saudi Arabia reported 8/400 (2%) over a 3 year period [51] but 9/90 (10%) of a series of children from Medina had a clinical stroke, with eight having an abnormal CT scan [52]. When adults are included, the prevalences of stroke in the Western, SouthWestern (7.5%) and Eastern (6%) provinces of Saudi Arabia are high [53,54,55,56].

### 2.5. Ischaemic Stroke

Overt ischaemic stroke is common in mid-childhood, between 2 and 10 years of age [2]. Hemiparesis is the typical presentation, and without secondary prevention, recurrence occurs in up 67% [27]. Ischaemic stroke incidence decreases to a minimum between the ages of 20 to 29 years, but there is a further peak after the age of 35.

#### 2.5.1. Incidence of Ischaemic Stroke

The Los Angeles cohort followed by Darleen Powars from 1965 to 1978, i.e., before the introduction of any preventative measures, reported an incidence under and over the age of 20 years of 0.761 and 0.524 per 100 PYO respectively [27] (Figure 1A). The overall age-adjusted incidence rate was 0.61 per 100 patient years of observation (PYO) for individuals with all forms of sickle cell disease (HbSS, HbSC, HbSβthalassemia) enrolled in the CSSCD from 1978 to 1988, with peak incidences in those aged 2–5 years (Figure 1A) and those >50 years [2].

In more recent USA studies of children with SCD, incidences of stroke of 0.13 to 0.51 [36] and 0.310 [25] per 100 PYO have been reported, while the rate for young adults was similar [25]. However, as more patients survive childhood, stroke has become relatively more important in adults with SCD and in the same study, those aged 35–64 years and >65 years, had rates of 1.16 and 4.7 per 100 PYO, about three times as high as adults in this age group in the general population [25].

The few available data [4] suggest that overall, the incidence of stroke in children with SCD residing in Africa is at least 0.88 per 100 PYO [57], equivalent to that seen in the USA before 2000 [2,27,58,59]. The incidence may be even higher, for example, 2.7 per 100 PYO in a study of 351 children in Ibadan, Nigeria, followed for 3 months to 4 years (personal communication I Lagunju), while the incidence was 3 per 100 PYO in another study of 66 children aged 37–197 months recruited over a period of 6 months and followed for 12 months [4,60] (Figure 1A).

#### 2.5.2. Effect of Interventions for Ischaemic Stroke on Incidence of Stroke in SCD

##### Newborn Screening and Prevention of Infection

The later the diagnosis, the higher the stroke incidence in children [31] so newborn screening and prevention of complications, including acute anaemia and infection, might prevent stroke. Meningitis secondary to *Streptococcus pneumoniae* and *Haemophilus influenzae* may be associated with cerebrovascular disease and contemporaneous or later stroke in the general paediatric population [61] but whether or not this has been an important cause in SCD is controversial [46]. Newborn screening for SCD began in some centres in the 1980s [59,62] and Penicillin prophylaxis for young children was introduced at around the same time [63], with vaccination against *Streptococcus pneumoniae* and *Haemophilus influenzae* following soon afterwards. These advances appeared to be associated with a reduction in mortality [59,62] and stroke incidence in children appeared to decrease in some regions of the USA and Europe during the 1980s and 1990s [24,37,62,64] (Figure 1A). However, despite newborn screening, the overall incidence in childhood in other areas was similar to that reported previously [2,27], with 0.67/100 PYO at the Children’s hospital of Philadelphia from 1990 to 1998 [65], 0.85/100 PYO in the Dallas cohort from 1983 to 2002 [59], 0.88/100 PYO in Northern California from 1991 to 1998 and 0.8/100 PYO in Nagpur, India from 2008 to 2012 [66] (Figure 1A).

##### Transcranial Doppler Ultrasound

Screening for stroke risk using TCD and appropriate management for those at high risk does appear to have successfully achieved primary stroke prevention. TCD is used to measure the time averaged mean of the maximum velocities (TAMMV) in the intracranial arteries (distal ICA, MCA, ACA and PCA), which are typically affected in SCD [67] (Figure 2). Increased TCD velocities may be associated with narrowing of the artery diameter [68] or increased CBF in the presence of anaemia [69]. A submandibular approach allows the detection of post-bulb extracranial carotid stenosis [70,71].

In SCD, stenosis on conventional arteriography may be detected when the distal ICA/MCA velocities are between 140 and 190 cm/s; velocities more than 190 cm/s are associated with marked artery stenosis on angiography [72]. Abnormal ICA/MCA TAMMV ≥ 200 cm/s are associated with a 40% risk of stroke over a three year period, while Conditional TAMMV (170–199 cm/s) are associated with a 7% risk over the same time period [73]. In the 1991–1998 East London study before treatment, the incidence of stroke in those with Abnormal TCD was 12.7/100 PYO (Figure 1B) and for those with Conditional TCD was 1.94/100 PYO. The incidence of stroke in those untreated with Abnormal TCD was 11.1/100 PYO in the Ibadan cohort [60] (Figure 1B).

##### Prevalence of Abnormal and Conditional Transcranial Doppler

Even in infancy ICA/MCA velocities are higher than control but those in the Conditional range are rare and Abnormal velocities have not so far been documented [74,75], so there are no data as yet on the range predicting stroke risk in this age group; most screening programmes start at age 2 years and stroke secondary to other mechanisms, e.g., embolus through a patent foramen ovale [76], may be important in those younger than this [6,7]. The prevalence of abnormal TCD in preschool and school-age children was 8% in the original Augusta cohort in the USA in the 1990s [73] and was 6% in the East London neonatally screened cohort in the UK [62]. A further analysis of TCD screening in 542 children in East London found that 18 (3%) had high abnormal TAMMV (>220 cm/s) [77]. An initial study in Paris found that 9.6% of children had abnormal TCD [78] and a subsequent analysis from their newborn cohort found that a quarter of children had abnormal TCD according to the Stroke Prevention Trial in Sickle Cell Anaemia (STOP) criteria [79]. Two of 48 (4%) of Spanish children with SCD had abnormal TCD [80]. Recent data from the 28-center DISPLACE consortium in the USA reported a lower rate of 2.9% with a median age for the first abnormal TCD of 6.3 years, but there were difficulties in implementation across sites and age at first TCD was typically only slightly lower than this [81]. The rate in Jamaica was reported as 6.7% but some of these children had prior stroke [82]. Of 178 patients screened in Nagpur, central India, 5 (3%) had abnormal and 8 (4.5%) had conditional TCD [83].

In the African meta-analysis, the prevalence of Abnormal and Conditional TCD was 6.1 and 10.6% respectively [4]. The rates of Abnormal and Conditional TAMMV were 8.4% and 21.9% respectively in Ibadan with repeat screening [84]. The rate of Abnormal TCD was 7% in an initial study in Tanzania [85] but subsequent studies found lower rates [86,87] consistent with those in other East African countries, including Kenya [88] and Uganda [33]. The rate of Conditional TAMMV was low in a study from the Arabian peninsula, including patients with SCD from Kuwait, Oman, United Arab Emirates and Iraq [89], with hydroxyurea appearing to have a beneficial effect. In 410 children living in the Gulf states, nearly 50% of whom were receiving hydroxyurea, only nine Conditional and one Abnormal TAMMV were documented using imaging TCD and the majority reverted to normal 2 months later [48]; normal TAMMV were more common in the Arab population with SCD than in those of Caucasian and African origin and in those who had a previous exchange transfusion.
jcm-10-04232-t001_Table 1Table 1Prevalence of Stroke and Conditional and Abnormal Transcranial Doppler in Africa.
Stroke PrevalenceTranscranial Doppler Prevalence Conditional/Abnormal
PopulationNumber Stroke (%)*n* StudiedConditionalAbnormalConditional + AbnormalAlgeria871 (1) [3]



Egypt1005 (5) [90]52

14 (30) [91]

785 (6) [92]
5 (6) [92]604 (7) [91]6010010 [93]Sudan

119000 [94]Mali

57268 (12)30 (5)98 (17) [95]Nigeria572171 (1) [96]



Enugu319523 (0.7) [97]



Lagos32217 (5) [46]



Ibadan50027 (5) [29]14529 (20)6 (4)35 (24) [98]Ibadan35124 (7) [31]26545 (17)19 (7)64 (18) [99]Ibadan2147 (3) [100]39674 (19)28 (7)102 (26) [101]Kano607 (12) [102]



Port





Harcourt25611 (4) [103]



South West2407 (3) [104]



Senegal4387 (2) [105]



Angola2006 (3) [106]



Cameroon1138 (7) [107]



1208 (7) [30]



967 (7) [108]32

12 (38) [109]Democratic Rep Congo50014 (3) [110]



Congo142214 (1) [111]



Kenya





Nairobi36012 (3) [112]



Kilifi1051 (1) [88]1050 (0)3 (3)3 (3) [88]Uganda2870147 (7) [113]



25615 (6) [33]25638 (15)5 (2)43 (17) [33]Tanzania





NorthWest12421 (17) [32]



Dar es Salaam20015 (8) [86]60125 (4)42 (7)67 (11) [85]

2001 (0.5)11 (5.5)12 (6) [86]

2243 (1)1 (0.5)4 (2) [87]Malawi11710 (9%) [114]





##### Effect of TCD Screening and Chronic Blood Transfusion on Risk of Stroke in Children with SCD and Abnormal TCD

The STOP study demonstrated that TCD is a useful tool for screening and detection of children with SCD at risk of stroke and that regular blood transfusion reduced risk [115], although for reasons that are poorly understood, there is substantial variation over time within individuals and TCD screening has not so far proved useful in screening adults with SCD for stroke risk. Children with TCD MCA or ICA TAMMV ≥ 200 cm/s are at high risk of first arterial ischaemic stroke (Figure 3A) over the subsequent few years (10–13% per year) if they do not receive indefinite regular blood transfusion therapy [68,73,115,116]. In the standard arm of the STOP trial, the incidence of stroke was 10.7 per 100 PYO, whereas in the transfusion arm, the stroke rate was 0.9 per 100 PYO [117] (Figure 1B). Those with Conditional TCD studies are also at higher risk of stroke than those with velocities < 170 cm/s [118].

Since the STOP trial ended prematurely because there was a very large advantage in favour of blood transfusion, screening and transfusion for those with velocities >200 cm/s has been recommended as standard care in the USA and the UK [119]. STOP2 showed that discontinuing transfusions led to a high rate of reversion to abnormal velocities and stroke which did not occur in the transfusion arm. The results from the landmark STOP and STOP2 trials formed the basis for the current gold standard of care for primary stroke prevention in SCD; yearly TCD screening with chronic blood transfusion therapy for children with CBFV >200 cm/s and close monitoring of those with conditional velocities (>170–200 cm/s). TCD screening and regular transfusion for those with abnormal TCD has apparently been associated with a substantially decreased incidence of stroke, with one centre reporting a reduction from 0.46 to 0.53 to 0.18 per 100 patient-years [64] and another reporting a decline from 0.67 to 0.06 per 100 patient-years [65] (Figure 1A). Epidemiological evidence also suggests that there has been a parallel fall in the overall incidence of stroke in sickle cell disease in the USA since TCD screening/chronic transfusions began from around 0.88 to 0.17–0.24 per 100 patient years [58,120] (Figure 1A), with a similar reduction in the UK to 0.27 per 100 patient years overall and 0.18 per 100 patient years for ischaemic stroke [62,77] (Figure 1A); this strategy appears to be cost effective [121].

##### Effect of Hydroxyurea on Risk of Stroke in Children with SCD and Abnormal TCD

Alongside the introduction of TCD, the increasing use of hydroxyurea in children with symptomatic SCD in the USA may have been one reason for the reduction in stroke incidence after 1998 [37]. The TCD With Transfusions Changing to Hydroxyurea (TWiTCH) study showed that for children with Abnormal TCD but normal magnetic resonance angiography (MRA) who had been transfused for a year, substituting hydroxyurea was not inferior to continuing chronic transfusions [122]. Subsequently a significant proportion of children with Abnormal TCD in the USA and Europe have switched, which raises the possibility that hydroxyurea might be a cost-effective alternative to transfusion for primary prevention of stroke in children with SCD and Abnormal TCD, particularly as the available data shows that hydroxyurea reduces TAMMV [119,123,124]. In a cohort study of 34 patients with abnormal TCD in Belgium treated with hydroxyurea 20 mg/kg/day, for those with abnormal TCD in the high stroke risk group, the incidence of stroke was 1 per 100 PYO [124], while in a feasibility single arm trial of fixed-dose 20 mg/kg hydroxyurea in northern Nigeria, the rate was 0.76 per 100 PYO [117]. The use of maximum tolerated dose in those with abnormal TCD in Ibadan reduced the overall incidence of stroke from 3 to 0.07/100 PYO or 0.08/100 PYO for those followed for more than 5 years [104] (Figure 1A), with incidences of 0.27/100 PYO in those with conditional or abnormal TCD treated with hydroxyurea and 0.5/100 PYO in those with abnormal TCD (Figure 1B). These figures for incidence are all less than the expected rate of 10.7 strokes per 100 PYO in the untreated arm of the STOP trial [117] (Figure 1B). In a randomised trial comparing 10 mg/kg and 20 mg/kg fixed dose hydroxyurea, the incidence rates of stroke were also both lower than the untreated arm of the STOP study and were not significantly different from each other [125,126].

TCD may detect cerebrovascular disease at an earlier stage than MRA; the highest risk of stroke is in children in whom both are abnormal, where cerebrovascular disease rarely improves without blood transfusion [127] and even with transfusion may not normalise completely [128]. Studies have shown a good correlation between abnormally high TCD velocities and Xenon CBF studies, conventional angiography [72] and MRA [127,129,130,131]. However, there are very few data on the natural history in the 2–7% of patients with abnormally low velocities [127,128,132] although they are relatively common in some populations and may be associated with vasculopathy on MRA [48,86].

##### Effect of Stem Cell Transplantation on Risk of Stroke in Children with SCD and Abnormal TCD

A nonrandomised controlled intervention study of matched sibling donor haematopoeitic stem cell transplantation compared with continuing transfusion for abnormal TCD found that the TAMMV was lower at 1 and 3 years in the transplanted group [133] and improvement in stenosis on MRA appears to be more common in those undergoing transplantation [134]. In the French preventative programme including immediate transfusion after one abnormal TCD, hydroxyurea and subsequent stem cell transplantation, no child with abnormal TCD at screening had a stroke [79].

### 2.6. Stroke Syndromes in Children with SCD and Normal TCD

In the East London cohort, for those with TCD classified as always normal, the incidence of stroke was 1.18/100 PYO. One teenager sustained white matter injury during an episode of posterior reversible encephalopathy syndrome after a chest crisis precipitated by appendicitis (Figure 3B), one child had an intracerebral haemorrhage during a painful crisis (Figure 3C), one child who had had a posterior TIA at the age of 4 years had an anterior stroke secondary to extracranial carotid disease aged 16 years (Figure 3D), one child had bilateral borderzone infarction in the context of a facial infection (Figure 3E) and one with haemoglobin SC disease and neonatal hydrocephalus had a straight sinus thrombosis with propagation and died of raised intracranial pressure (Figure 3F).

### 2.7. Intracranial Haemorrhage

Intraparenchymal, intraventricular, subarachnoid and occasionally subdural haemorrhages have all been described in patients with SCD [135]. Haemorrhage has the highest incidence in young adults (20–30 years) but is not uncommon in children [2]. Patients with prior infarction are at increased risk of haemorrhage as they age [27]. Subarachnoid and intracerebral haemorrhage occur in the context of acute hypertension and may be associated with corticosteroid use, recent transfusion or bone marrow transplantation [135]. Cerebral haemorrhage in SCD is commonly related to aneurysm formation even in children [39]. The aneurysms which rupture are typically located at the bifurcations of major vessels, particularly in the vertebrobasilar circulation [39]. Intraparenchymal bleeding may be associated with large-vessel vasculopathy, especially if moyamoya formation is present [136]; in fact stenotic vasculopathy was found in five of seven children with SCD and haemorrhagic stroke in one series [39]. Venous sinus thrombosis [137] and reversible posterior leukencephalopathy [15] may also be associated with haemorrhage. There are reports of epidural haematomata in the absence of significant head trauma in SCD, probably related to hypervascular areas of bone [138].

In the CSSCD, the incidence of haemorrhagic stroke was 0.15, 0.25 and 0.14 per 100 PYO between ages 2 and 5, 6 and 9 and 10 and 19 years respectively but increased to 0.44 per 100 patient years between ages 20–29 years, i.e., the incidence was higher during the age range for which risk of ischaemic stroke had fallen, potentially indicating different underlying mechanisms or progressive vasculopathy [2]. Even in children the mean age at presentation is higher in haemorrhagic than ischaemic stroke [39]. It is not yet clear whether the incidence of haemorrhagic stroke has been altered by interventions for children with SCD. However, the available data suggests that it has not as for patients in the East London network screened with TCD between 2001 and 2012 and transfused for Abnormal TCD, there were 0.09 haemorrhagic strokes per 100 PYO [77] when the previous rate in that cohort undergoing TCD screening between 1991 and 2000 but not treated was 0.07 per 100 PYO [38] and the rate in the 1988–1991 Baltimore study was 0.048 per 100 PYO [24].

### 2.8. Stroke Recurrence

In untreated patients, the risk of recurrence ranges from 60 to 92% in patients with sickle cell anaemia [27,28,29,30,139,140,141,142,143]. A recent study from Nigeria reported an incidence of 17.4 per PYO for a recurrent stroke after an initial event [57]. With regular blood transfusion, around 10% of patients experience a recurrence [144], with an incidence of 0.66–1.6/100 PYO [145,146]. Recurrence is more common in those whose stroke did not occur in the context of an acute illness [147] and if moyamoya collaterals are demonstrated angiographically [136,148]. The obvious reduction in risk documented in the early studies [27,141,143] meant that randomised controlled trials of blood transfusion to prevent clinical stroke recurrence in SCD were never considered ethical but data are available for those with SCI. In the observational CSSCD, compared with the stroke incidence of 0.54/100 PYO for the whole cohort, children with SCI on MRI had nearly double the incidence of clinical stroke to 1.03/100 PYO, with an additional incidence of 7.06/100 PYO for new or more extensive SCI [149]. A randomised controlled trial of regular blood transfusion for 3 years compared with standard care for children with SCI on MRI found that that the incidence was respectively 2.0 and 4.8 events per 100 years for the primary end point (overt stroke or recurrent SCI) [150]. If blood transfusion is not feasible, hydroxyurea also appears to reduce the risk of recurrent stroke [151,152,153] although a randomised trial in the USA appeared to demonstrate inferiority compared with regular blood transfusion [154].

## 3. Seizures

Patients with SCD may also have single and recurrent seizures [12,45,46,155,156]. Most of the cohort data comes from low-middle income countries where neuroimaging is unaffordable, making the distinction between febrile and acute symptomatic seizures in young children very difficult. Between 7% and 10% of individuals with SCD will experience at least one seizure [12,45]. In the CSSCD, 5.5% of children and 9% of adults had seizures with a median ages at onset of 8.5 and 28.0 years respectively [155]; CVA and meningitis were the most common aetiological factors in both age groups while some may also have had posterior reversible encephalopathy syndrome in the context of pneumonia, acute anaemia or nephrotic syndrome. A case control study from Lagos, Nigeria found that 10% of children with SCD had febrile seizures compared with 2% of controls [46]. In the Jamaican cohort of 543 SCD patients of all genotypes, febrile seizures were documented in 16, acute symptomatic seizures in 14, single unprovoked seizures in 5 and epilepsy in 12 [12]. Febrile seizures (HbSS *n* = 6, HbSC *n* = 1), acute symptomatic seizures (*n* = 5), single unprovoked seizures (*n* = 2) and epilepsy (separate occurrence of two or more unprovoked seizures; *n* = 6) were all diagnosed in Lagunju’s description of adverse neurological outcomes in 214 Nigerian children with SCD [45]. In the Jamaican Cohort Study of Sickle Cell Disease, the incidence of epilepsy was 1/100 PYO for all genotypes and 1.39/100 PYO for those with HbSS, making this diagnosis 2–3 times more common than in the general population [12]. The five-year cumulative incidence of febrile convulsions was 2.2% in this study [12]. In a meta-analysis of studies from Africa, the prevalence of seizures was 4.4% [4] and was higher in studies with a smaller sample size and in more recent studies.

## 4. Headache

Headache affects between 20% and 45% of patients with sickle cell disease and may occur at any age, including in young children [4,10,157,158,159]. A case control study from Lagos, Nigeria found that 25% of children with SCD aged 4–14 years had headaches compared with 7.3% of healthy controls with HbAA recruited from a local school [46]; the proportion in the SCD population stayed the same in adolescents and adults, while that in controls recruited from the university increased in adolescence, although the difference remained statistically significant at all ages. Severe headache can also be a symptom of intracranial haemorrhage: subdural, intraparenchymal, subarachnoid or intraventricular [160] and neuroimaging should be performed as an emergency at initial presentation. Venous sinus thrombosis [137,161,162,163] and pseudotumor cerebri [164,165] have also been reported and should be excluded in those presenting with acute headache.

## 5. Central Nervous System Infections

In addition, children with SCD can present with central nervous system infections such as meningitis, bacterial abscess and cerebral tuberculoma. The incidence of CNS infections has decreased with penicillin prophylaxis and immunisation in the USA [160,166] but these preventative strategies are not widely available in Africa where these conditions are still prevalent alongside malaria.

## 6. Coma

Coma in SCD may be due to intracranial haemorrhage (Figure 3C) [135,160], although extensive middle cerebral artery infarction with oedema and midline shift (Figure 3A) [167], posterior reversible encephalopathy syndrome (Figure 3B), which may progress to borderzone infarction (Figure 3E), and venous sinus thrombosis (Figure 3F) [137] can also have a similar presentation. Other symptoms of covert (‘silent’) cerebral infarction include dysphasia, difficulty with gait (Figure 3D) [168] and ‘soft neurological signs’ [160,169,170,171].

## 7. Other Neurological Presentations

There are clinical descriptions of myopathy [172], myelopathy and neuropathy [160,166,173,174,175,176,177] but access to a neurological opinion and investigation with electromyography and nerve conduction and/or muscle biopsy may not have been available. There are few population-based studies of prevalence but one case-control study in Nigeria found a high prevalence of sensory neuropathy [46]. This study also found a trend for an excess of tremor [46], which might be related to basal ganglia ischaemia.

## 8. Cognitive Impairment

Studies rarely report formal diagnoses of cognitive impairment, which makes meaningful comment on the prevalence challenging. The overwhelming majority of studies also report sample-level mean performance on cognitive tests, rather than the proportion of patients with scores that fall into established clinical categories. In single centre studies from the USA and UK, up to 50% of very young children scored below the average on developmental scales [74,178,179]. The large French multicentre study found that 9% of controls and 30% of children with SCD had an IQ < 75 [78]. An Italian study of children with SCD, almost all of whom spoke two or three languages, found that a quarter had a full scale IQ < 75, while three quarters had a lower verbal than performance IQ [180]. In a UK study, 28% of children, adolescents and adults had processing speed scores that fell in the borderline to extremely low categories (i.e., scores of < 80) compared to 6% of controls and when this was accounted for, full scale IQ was not different between the groups [181]. A study from Cameroon found that 37.5% of children with SCD had mild to severe cognitive difficulties [108] and the proportion was higher in a Tanzanian study with controls [182]; the results of tests developed outside Africa should be interpreted with caution although understanding the risk factors for cognitive impairment is likely to be important in the development of treatment. A recent study using Patient Report Outcome Measures found that more than 50% of adolescents and adults reported difficulties in domains of attention, executive functioning, processing speed, and reading comprehension [183]. Although there is significant variability within SCD populations, patients are at greater risk of clinically significant cognitive impairment than matched controls. Neurodevelopmental screening using questionnaires and relatively simple tools is recommended in the recent guidelines, particularly in early childhood [119].

## 9. Abnormalities Detectable on Cross-Sectional Neuroimaging in Asymptomatic Patients

### 9.1. Covert (Silent) Cerebral Infarction

Silent cerebral infarction is found in up 40% of sickle cell patients without clinical symptoms overall [21], including those with sickle β-thalassaemia [184], with a steady rate of accumulation to >50% by in adults with a median age of 30 years (interquartile range 22–35 years) [158,185,186]. In the CSSCD, SCI was associated with a 14-fold increase in risk of ischaemic stroke, and 25% of children with SCI presented with new or enlarged lesions at follow-up [149]. After symptomatic stroke SCI may progress despite regular transfusion [187].

SCI may occur as early as the 6th month of life [188,189]. There is evidence from the USA and Europe that prevalence reaches 25% by 6 years of age [190], 39% by 18 years of age [191] and 53% by young adulthood [185], with no reports of a plateau and more than one lesion in 37% of patients with SCD. In Tanzanian studies of children with TAMMV outside [86] and within [87] the normal range on TCD screening, 43% and 27% respectively had SCI. Of note, prevalence estimates may vary not only with age but also with scanner magnet strength and voxel size [7]. The high prevalence of silent infarcts on MRI in children with SCD compared with children with non-sickle stroke [192] probably at least in part reflects the chronicity of the vascular compromise in this population. Silent infarction may be associated with intracranial cerebrovascular disease, such as stenosis or moyamoya syndrome [193,194] but many patients with SCI have normal TCD and MRA and alternative mechanisms include embolus associated with right-to-left shunting [195].

Patients with SCD and overt stroke or SCI may be more likely to have abnormal psychometric testing [196,197]. However, MRI studies at higher field strength have been less likely to find an association between cognitive function and SCI [182,198,199]. Quantitative MRI may clarify some of these issues [7] as well as providing information of clinical use in addition to an experienced radiologist’s reading of the conventional MRI study.

In the CSSCD, silent cerebral infarction was also associated with cognitive decline [200] These findings have been replicated in more recent work, including in a study where silent cerebral infarction in patients younger than 5 years old were shown to be associated with later progressive ischaemia, vasculopathy, academic difficulties and a higher risk of stroke [189]. Another study found no evidence of decline unless there was moyamoya [201], but the sibling controls improved over time whereas the children with SCD did not [202].

### 9.2. Cerebrovascular Disease on Magnetic Resonance Angiography

In sickle cell disease, MRA can be up to 85% accurate when compared with conventional angiography [203]. Turbulence or signal dropout on MRA may be graded as mild, moderate or severe [87,204], but there are few data looking at the relationship with the degree of arterial stenosis on conventional angiography [9,131]. MRA can detect cerebrovascular disease in very young children. In one study [188], MRA abnormalities were found in 3 out of 29 patients from 7 to 48 months of age, although there were none with MRA abnormalities in the 23 studied at baseline for the baby HUG trial [205].

In the STOP randomised trial of patients with abnormal TCD, MRA was undertaken at baseline in 100 patients, 47 in the transfusion arm and 53 in the standard care arm and was normal in 75 patients, while 25 demonstrated stenosis, mild in 4 and severe in 21 [127]. In the standard care arm, 4 of 13 patients with abnormal MRA findings had strokes compared with 5 of 40 patients with normal MRA findings (*p* = 0.03) [127].

Ectasia of the basilar and intracranial circulations has also been documented and is associated with low haematocrit [206,207]. Extracranial ICA stenosis and occlusion in children with sickle cell anaemia have been reported [208]; pathogenesis is unclear but some may be secondary to extracranial dissection while others appear to be associated with intracranial stenosis [194]. There is controversy over whether extracranial stenosis is associated with SCI [191,194]. Extracranial carotid artery occlusion or dissection should be considered in children with sickle cell anaemia presenting with symptoms of stroke and imaging of the neck vessels should be part of the investigation of these patients in the acute phase [208].

## 10. Risk Factors for Neurological, Cognitive and Neuroradiological Abnormalities

The various neurological and cognitive complications of SCD share some risk factors but apparently not others, although studies have been relatively limited in size and geography [209]. The phenotype may be influenced by the different β-haplotypes (the nucleotide 5′and 3′ sickle cell gene sequence). There are three major African and African American haplotypes: Senegal, Benin and Bantu (or Central African Republic) [210]. In addition, there is an independent haplotype in India and Saudi Arabia [211]. However, the majority of the available data suggests that the β-haplotypes are not associated with stroke [93,212,213,214,215] and, although there have been few studies in Africa, there is currently no evidence that there are large difference in prevalence of stroke between populations [4,47]. Stroke risk is reduced in the presence of co-inherited alpha-thalassemia [85,215,216]. The effect of co-inherited Glucose-6-phosphate dehydrogenase deficiency remains controversial [85,193,217,218,219,220,221]. Combinations of these variants and other single nucleotide pulymorphisms may be important factors in the development of vasculopathy in children with SCD [39,222].

In addition to abnormal TCD in cohort studies [209], the biggest risk factor for stroke in population studies of children and adults with SCD is hypertension [25,223], with diabetes mellitus, hyperlipidemia, atrial fibrillation and renal disease also risk factors in adults [25]. Predictors of ICA/MCA velocity and/or abnormal TCD [209] include low haemoglobin [82,84,224], haematocrit [225], haemoglobin oxygen saturation [82,84,88,226] and markers of haemolysis including reticulocyte count [82], aspartate transaminase [224] and lactate dehydrogenase [227]. Associations with poor cognitive function and school performance include low body mass index and overnight oxygen desaturation [80]

## 11. Conclusions

Although in SCD the incidence of first clinical stroke has fallen over the past 3–4 decades in the USA and Europe, this is still a highly prevalent problem in Africa, where the majority of the people with this condition live and where resources for screening and preventative treatment remain scarce. Evidence-based prevention of haemorrhagic stroke at any age and ischaemic stroke in adults is not yet feasible. Further epidemiological studies may allow cost-effective prevention of stroke and other neurological problems across the world.

## Figures and Tables

**Figure 1 jcm-10-04232-f001:**
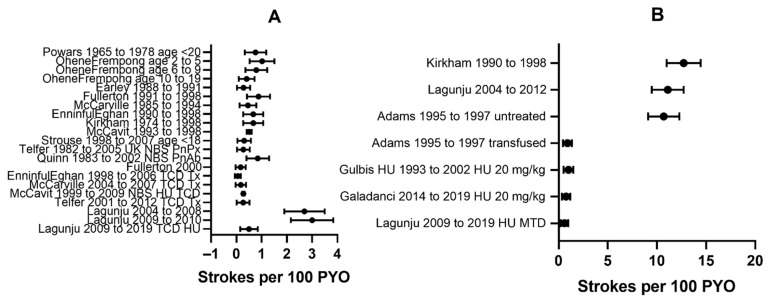
Incidence of Stroke in unscreened children with sickle cell disease (**A**) and in children with SCD with abnormal Transcranial Doppler (**B**). NBS Newborn screening PnPx Pneumococcal prophylaxis with Penicillin and vaccination PnAb Pneumococcal prophylaxis with Penicillin PC Personal Communication TCD Transcranial Doppler screening and treatment. HU Hydroxyurea. MTD Maximum tolerated dose.

**Figure 2 jcm-10-04232-f002:**
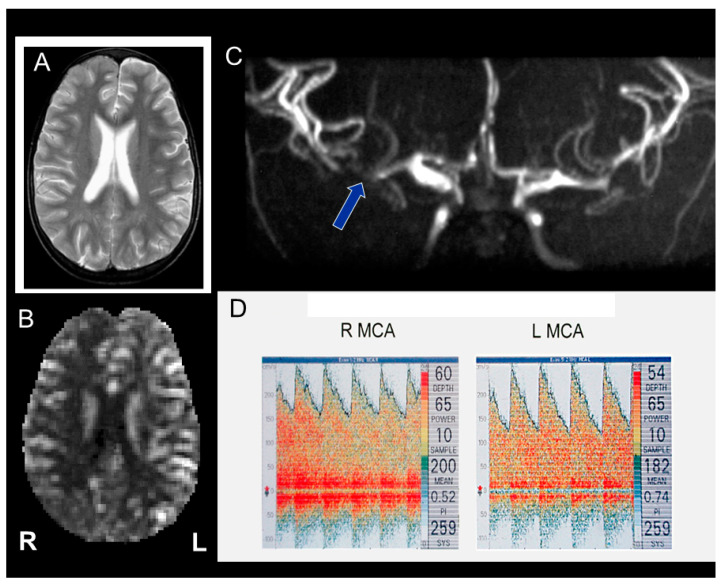
Magnetic resonance imaging (MRI) and transcranial Doppler) (TCD) in an 11 year old child with headache after a chest crisis. (**A**) T2-weighted MRI was normal but (**B**) there was reduced cerebral blood flow throughout the right hemisphere and posteriorly on the left. (**C**) Magnetic resonance angiography (MRA) showed signal dropout in the middle cerebral arteries, worse on the right, which manifest as (**D**) Abnormal and Conditional time averaged mean of the maximum velocity on the right and left respectively. He was transfused for a year and both MRI and TCD returned to normal.

**Figure 3 jcm-10-04232-f003:**
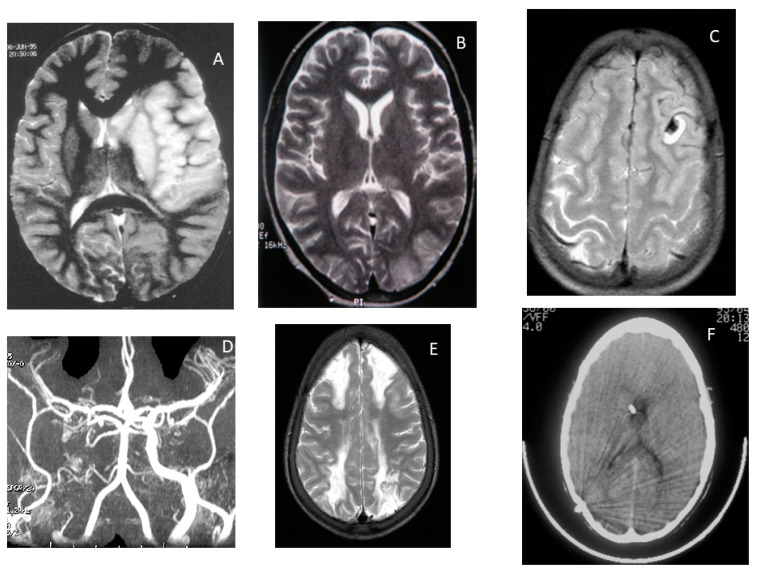
Stroke syndromes in sickle cell disease (**A**) arterial ischaemic stroke secondary to middle cerebral artery vasculopathy (**B**) posterior reversible encephalopathy syndrome (**C**) spontaneous intracerebral haemorrhage (**D**) extracranial carotid occlusion in a teenager with previous ataxia presenting with stroke (**E**) bilateral watershed infarction (**F**) straight sinus thrombosis.

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
