# Peer review of "Epidemiology of Stroke in Sickle Cell Disease"

_jcm, 2021, doi:10.3390/jcm10184232_

Round 1

Reviewer 1 Report

Thank you for this nice overview of stroke in sickle cell disease.

Good article to focus again on the importance of following up the risk of stroke not only in children but as well in adults since a lot of our colleagues in adult hematology are not familiar with this progressive disease in our developped countries.

Also very nice overview of the prevalence of stroke in Africa. 

Maybe a little bit to long if you could shorten the text by 25% the message would be even stronger .

Thank you!

Author Response

We thank the reviewer for the the positive comments

We were asked by the other reviewers to add small sections of text. We could cut down by including the non African prevalence data as a table but this would take time

Reviewer 2 Report

The authors perform a review on a very important topic in sickle cell disease. The review highlights the major points in sickle cell disease cerebral vasculopathy and stroke primarily, is well written and clear.

In some paragraphs, some issues could be better highlighted and a few references should be added. See my comments below:

1) Paragraph "7.Cognitive Impairment ": this is a very important issue that deserves more explanation. Although it is true that there is difficulty in comparing, as the authors state many publications do not report "the proportion of patients with scores that fall into established clinical categories". They should cite some of the references that do indeed report this and/or from different settings like: a) Bernaudin F, Verlhac S, Fréard F, Roudot-Thoraval F, Benkerrou M, Thuret I, Mardini R, Vannier JP, Ploix E, Romero M, Cassé-Perrot C, Helly M, Gillard E, Sebag G, Kchouk H, Pracros JP, Finck B, Dacher JN, Ickowicz V, Raybaud C, Poncet M, Lesprit E, Reinert PH, Brugières P. Multicenter prospective study of children with sickle cell disease: radiographic and psychometric correlation. J Child Neurol. 2000 May;15(5):333-43.

b) Montanaro M, Colombatti R, Pugliese M, Migliozzi C, Zani F, Guerzoni ME, Manoli S, Manara R, Meneghetti G, Rampazzo P, Cavalleri F, Giordan M, Paolucci P, Basso G, Palazzi G, Sainati L Intellectual function evaluation of first generation immigrant children with sickle cell disease: the role of language and sociodemographic factors.  Ital J Pediatr. 2013 Jun 4;39:36.

2) Paragraph "9.Risk factors for neurological, cognitive and neuroradiological abnormalities ". Several studies exploring SNPs have been performed in different settings (Brazil and Italy): although not definitive the possibility that SNPs might influence stroke risk and large vessel vasculopathy should be added: a) Belisário AR, Nogueira FL, Rodrigues RS, Toledo NE, Cattabriga AL, Velloso-Rodrigues C, Duarte FO, Silva CM, Viana MB. Association of alpha-thalassemia, TNF-alpha (-308G>A) and VCAM-1 (c.1238G>C) gene polymorphisms with cerebrovascular disease in a newborn cohort of 411 children with sickle cell anemia. Blood Cells Mol Dis. 2015 Jan;54(1):44-50.

b) Martella M, Quaglia N, Frigo AC, Basso G, Colombatti R, Sainati L. Association between a combination of single nucleotide polymorphisms and large essel cerebral vasculopathy in African children with sickle cell disease.Blood Cells Mol Dis. 2016 Oct;61:1-3.

Author Response

We thank the reviewer for their comments and suggestions:

1) We apologise for this oversight and have added sentences and the suggested references

2) We have added a sentence of combinations of SNPs with the suggested references

Reviewer 3 Report

I read with interest this detailed and well-written manuscript on the epidemiology of stroke in sickle cell disease. Generally, the scope of literature included in this study is adequate. The authors give a good review of this very important topic, which will substantially improve clinical practice and guide future research. I have a few recommendations and questions:

MAJOR ISSUES

  1. The authors state that stroke is a well-recognised complication of homozygous sickle cell anemia and occurs, albeit less commonly, in compound heterozygotes, e.g. those with HbSC disease and HbS-beta thalassaemia (HbSβthal). This is at least debatable. Based on exiting reports in the public literature, complications in SCD including stroke may not show substantial variations between SCD phenotypes/genotypes. In this article, the authors cite one such study (https://doi.org/10.1111/j.1600-0609.1993.tb00613.x). The authors should consider revising this statement and other sections based on this assumption.
  2. Conspicuously missing in the transfusion/stroke recurrence section(s) is the role of regular blood-transfusion therapy on the recurrence of stroke. In lines 335-336, the authors cite an older study that reviewed the clinical course and experience with transfusion therapy and compared findings to reports in the existing literature (DOI: 10.1016/s0022-3476(95)70204-0). More recently, controlled trials have been conducted to provide further clarity (e.g. DOI: 1056/NEJMoa1401731). The authors should consider integrating the current evidence in the review, due to the potential clinical relevance.
  3. The authors state that “In childhood approximately 75% of the infarcts are ischaemic and the remainder are haemorrhagic. This is based on the report by Powars et al 1. Subsequently, other studies have cited different rates (e.g. https://doi.org/10.1093/qjmed/hcy066). Basing findings on a single study/report is limiting, especially when multiple reports currently exist.
  4. To guide clinicians and researchers, this review might benefit from a line or two on the limitations of TCD. For example, the utility of TCD screening in individuals aged >16 years has not been established in controlled trials. As demonstrated in the STOP cohort, and other longitudinal studies, TCD velocities may also substantially vary over time for the same individual.
  5. Following observations including a higher rate of strokes when transfusion therapy is discontinued, some trials have evaluated the role of hydroxyurea as an alternative to indefinite transfusions (DOI: 10.1016/S0140-6736(15)01041-7). A section or paragraph summarizing the role of hydroxyurea following chronic transfusion might be valuable.
  6. Authors should define abbreviations when it is used for the first time. An example is the Stroke Prevention Trial in Sickle Cell Anemia (STOP)
  7. In some sections of the manuscript, it remains unclear whether the authors talk about the general population or individuals with SCD.
  8. The use of infarcts to represent strokes (e.g. line 27) might be problematic. The authors should consider being consistent with their terminologies.
  9. Lines 90-92: The sentence doesn’t read well. The authors should consider checking and rephrasing/insert punctuations
  10. Lines 198-199 Sentence appears unclear
  11. Is line 275 starting a new sub-section?

Author Response

  1. The sentence has been modified and the reference has been cited earlier
  2. We appreciate the reviewer having picked up this oversight. The text has been modified to include more discussion of the effect of blood transfusion and the RCT has been cited
  3. Additional text has been added to this section with appropriate recent references from McCavit, Baker and Kossorotoff, while we have moved a sentence on our data further up and added a sentence later on McCavit's data.
  4. These points have now been addressed at the beginning of the TCD section
  5. A sentence on the TWiTCH trial has been added
  6. STOP has been fully spelt out
  7. We have added "with SCD"
  8. We have reworded that sentence in the abstract
  9. The word "in" has been added
  10. We have added "e.g. embolism" for clarification
  11. Yes and the heading is now separate